# “Head-to-Toe” Lipid Properties Govern the Binding and Cargo Transfer of High-Density Lipoprotein

**DOI:** 10.3390/membranes14120261

**Published:** 2024-12-06

**Authors:** Florian Weber, Markus Axmann, Erdinc Sezgin, Mariana Amaro, Taras Sych, Armin Hochreiner, Martin Hof, Gerhard J. Schütz, Herbert Stangl, Birgit Plochberger

**Affiliations:** 1Department of Medical Engineering, Upper Austria University of Applied Sciences, 4020 Linz, Austriamarkus.axmann@fh-linz.at (M.A.); armin.hochreiner@fh-linz.at (A.H.); 2Science for Life Laboratory, Department of Women’s and Children’s Health, Karolinska Institutet,171 77 Solna, Sweden; erdinc.sezgin@ki.se (E.S.); taras.sych@ki.se (T.S.); 3J. Heyrovský Institute of Physical Chemistry of the Czech Academy of Science, 182 00 Prague, Czech Republic; mariana.amaro@jh-inst.cas.cz (M.A.); martin.hof@jh-inst.cas.cz (M.H.); 4Institute of Applied Physics, TU Wien, 1040 Vienna, Austria; gerhard.schuetz@tuwien.ac.at; 5Center for Pathobiochemistry and Genetics, Institute of Medical Chemistry, Medical University of Vienna, 1090 Vienna, Austria; 6Research Group Nanoscopy, Ludwig Boltzmann Institute for Experimental and Clinical Traumatology, 1200 Vienna, Austria

**Keywords:** lipoprotein, membrane order, Laurdan polarity, hydrogen bond network, glycerol region mobility

## Abstract

The viscoelastic properties of biological membranes are crucial in controlling cellular functions and are determined primarily by the lipids’ composition and structure. This work studies these properties by varying the structure of the constituting lipids in order to influence their interaction with high-density lipoprotein (HDL) particles. Various fluorescence-based techniques were applied to study lipid domains, membrane order, and the overall lateral as well as the molecule–internal glycerol region mobility in HDL–membrane interactions (i.e., binding and/or cargo transfer). The analysis of interactions with HDL particles and various lipid phases revealed that both fully fluid and some gel-phase lipids preferentially interact with HDL particles, although differences were observed in protein binding and cargo exchange. Both interactions were reduced with ordered lipid mixtures containing cholesterol. To investigate the mechanism, membranes were prepared from single-lipid components, enabling step-by-step modification of the lipid building blocks. On a biophysical level, the different mixtures displayed varying stiffness, fluidity, and hydrogen bond network changes. Increased glycerol mobility and a strengthened hydrogen bond network enhanced anchoring interactions, while fluid membranes with a reduced water network facilitated cargo transfer. In summary, the data indicate that different lipid classes are involved depending on the type of interaction, whether anchoring or cargo transfer.

## 1. Introduction

The structure and organization of cell membranes are of significant interest, particularly in the context of lipid–protein interactions. In addition to maintaining membrane integrity and acting as signaling molecules, lipids play a structural role in shaping physical membrane properties. Membranes can exhibit multiple possible phase states and their properties are determined by the constituting lipid structure [1]. Moreover, membrane properties such as lateral fluidity strongly depend on temperature. At sufficiently low temperatures, the solid-like gel [2] phase is characterized by low lateral lipid motion [3], high lipid packing/membrane order, and a low level of membrane hydration [4]. Above the phase transition temperature (i.e., at the “melting” temperature *T_m_*), lipids form the so-called liquid-disordered L_d_ (also known as fluid, liquid–crystalline, or lamellar L_α_) phase. The *T_m_* value strongly depends, among other parameters, on the saturation of the fatty acid chains. In the presence of cholesterol (Chol) or similar sterols another phase emerges, known as the liquid-ordered L_o_ phase [5]. Compared to gel phases, the L_o_ and L_d_ phases demonstrate lipid diffusion, though in the case of L_o_ phases, this diffusion is reduced due to the lipid composition [6,7,8,9]. Furthermore, it also exhibits similarities to the gel phase with regard to membrane order and lipid packing. The L_d_ and L_o_ phases can coexist, and their size and lifetime are variable and temperature-dependent [1,10]. Their inherent stability (i.e., not merging into a single, larger domain) depends on a high-energy barrier due to domain curvature at the phase boundaries [11].

Glycerophospholipids, where acyl fatty acid chains are attached to a glycerol backbone by ester or ether bonds, are the main lipid constituent of biological membranes. Ether lipids are a substantial structural component in cell membranes, representing around 20% of the total phospholipid content in mammals [12]. Their most significant contribution is to the membranes of the brain, heart, kidneys, lungs, skeletal muscle, and immune system [13]. Plasmalogens, the most abundant form of ether lipids, distinguished by a cis double bond adjacent to the ether linkage, are notably concentrated in the Chol-rich regions of membranes [14,15]. They play a crucial role in facilitating cellular Chol efflux [16]. Notably, the liver contains a very low amount of ether lipids. Comprising Chol esters and triglycerides surrounded by a shell of phospholipids, Chol, and proteins, high-density lipoprotein (HDL) particles facilitate the transport and transfer of non-water-soluble substances to and from cells. HDL particles, which are secreted by the liver and contain approximately 20–30% plasmalogens relative to their total glycerophospholipid pool, transport plasmalogens to other tissues, thus depleting the liver of ether lipids [17]. Cholesterol is an essential building block, comprising up to 40 mol% [18] and, in certain specialized membrane environments, reaching up to 50% [19] in eukaryotic plasma membranes, where it plays a crucial role in regulating membrane fluidity. Notably, Chol enrichment slows down ether lipid uptake [20].

In addition to the diversity of lipids and their composition, the interfaces of lipid membranes create an equally complex environment and play a key role in the dynamics, stability, and function of biological membranes. These interfaces are crucial for important cellular processes such as protein binding [21] or drug interaction [22,23]. The membrane interface is characterized by an abundant network of dynamic hydrogen-bonded water chains bridging the lipid headgroups, some of which form transient lipid clusters [24]. Numerous molecular dynamics simulations [25,26] have been conducted to investigate how water molecules interact with anionic lipids at membrane interfaces and how they are involved in protein binding. Membranes differ in properties such as viscoelastic and compressibility parameters. A wide spectrum of techniques and lipid compositions are applied in an attempt to provide a comprehensive view of these individual segments and how they affect each other and their vicinity [27,28,29]. The use of environmentally sensitive fluorescence probes like Laurdan is beneficial for studies of biological heterogeneity, in particular within the glycerol region of lipids. Laurdan is sensitive to the polarity and viscosity of its local environment [30]. Thus, Laurdan makes it possible to quantify the hindrance of lipid internal dynamics, so-called glycerol region mobility. Changes in membrane properties, most notably its lipid packing, cause shifts in Laurdan’s emission spectrum, which are quantified by the generalized polarization (GP) value. This spectrum change can be influenced by the hydration state, in part, but also by Chol [28] due to an altered local hydrogen bond network. Time-dependent fluorescence shift (TDFS) measurements allow for the mapping of the interfacial region, i.e., the local environment of the lipid’s glycerol region, characterizing both the dynamics of the hydrated segments of lipid membranes as well as the level of its hydrogen bond network. Hence, this makes it possible to quantitatively characterize the mobility of hydrated lipid segments that are in close vicinity of the probe fluorophore [31]. Imaging techniques such as single-molecule fluorescence microscopy (SMFM) and fluorescence correlation spectroscopy (FCS) were used to visualize different lipid phases or to determine the motion of biomolecules [32,33]. Phase-specific fluorophores like DiI [34] enable visualizing and characterizing the lipid bilayer’s spatial organization. To localize Chol in membrane phases or to monitor sterol uptake and flux in cells [35], Cholesterol-BodipyFL (Chol-BodipyFL) is predominantly adopted. It is a well-established Chol analog with a conjugated boron–dipyrromethene difluoride fluorophore. The partition coefficient (*K_P_*) can be used to describe quantitative partitioning in phase-separated membranes [36]. Previous studies [37] have demonstrated that HDL particles interact directly with the membrane without any receptor involvement. Lipoprotein particles showed strong inhibition in their interaction with artificial lipid membranes as their cholesterol content increased. This study assesses the interaction of HDL particles with the membrane influenced by (i) the glycerol linkage of the fatty acid chains to the head group (i.e., ester vs. ether lipids), (ii) the Chol content of the target membrane, and (iii) the saturation and length of the fatty acid chains using multi-correlative methods, including FCS, TDFS, and SMFM.

## 2. Materials and Methods

### 2.1. Materials

Sephadex G-25 fine resin, potassium bromide, sodium chloride, calcium chloride, EthyleneDiamineTetraacetic Acid (EDTA), Tris-(hydroxymethyl)-aminomethan-hydrochlorid (Tris/HCl), 1× Phosphate-Buffered Saline (PBS), 2-(4-(2-HydroxyEthyl)-1-PiperazineEthaneSulfonic acid (HEPES), Nunc^®^ Lab-Tek^®^ Chamber Slides, 1,2-DiOleoyl-sn-glycero-3-PhosphoCholine (DOPC (18:1)), 1,2-DiPalmitoyl-sn-glycero-3-PhosphoCholine (DPPC (16:0)), 1-Palmitoyl-2-Oleoyl-sn-glycero-3-PhosphoCholine (POPC (16:0-18:1)), 1,2-Di-O-hexadecyl-sn-glycero-3-PhosphoCholine (DietherPC (16:0)), 1,2-Di-O-octadecyl-sn-glycero-3-PhosphoCholine (DietherPC (18:0)), 1,2-Di-O-(9Z-octadecenyl)-sn-glycero-3-PhosphoCholine (DietherPC (18:1)), N-(octadecanoyl)-sphing-4-enine-1-phosphocholine (also known as. brain SphingoMyelin (bSM)), Chol, and Chol linked to boron dipyrromethene difluoride at sterol carbon-24 (Chol-BodipyFL) were purchased from Merck, Vienna, Austria. 1,2-DiPalmitoyl-sn-glycero-3-PhosphoEthanolamine linked to abberior STAR RED (DPPE-ASR) was purchased from abberior, Göttingen, Germany. Atto 647 NHS Ester, 1,1′-Dioctadecyl-3,3,3′,3′-tetramethylindocarbocyanine perchlorate (also known as DiI™), Laurdan [38] were purchased from Thermo Fisher Scientific, Vienna, Austria.

### 2.2. HDL Particle Isolation and Labeling

Lipoprotein particles were isolated as previously described [39]. Lipoprotein isolation was approved by the Ethics Committee, Medical University of Vienna (EK-Nr. 1414/2016). Written informed consent was obtained from all participants. Briefly, plasma was recovered from whole blood obtained from normolipidemic healthy volunteers by centrifugation (twice at 3000× *g*, 4 °C, 20 min) and its density was adjusted to 1.063 g/L using KBr. The samples were centrifuged (52,000× *g*, 4 °C, 20 h) and the upper phase containing VLDL and LDL was discarded. The density of the bottom fraction was adjusted to 1.21 g/L with KBr and the samples were centrifuged as described above. The upper phase containing HDL particles was recovered and was re-adjusted to a density of 1.21 g/L and centrifuged a second time to ensure complete removal of serum albumin. The sample containing HDL particles was dialyzed extensively against 0.9% (*w*/*v*) NaCl and 0.1% (*w*/*v*) EDTA, pH 7.4, to remove KBr. Protein concentration was determined by the Bradford method. The (apo-lipo)protein(s) of HDL particles were labeled with Atto 647 via NHS-based covalent linking according to the manufacturer’s protocol. Briefly, the labeling procedure includes conjugation of the dye to amino groups at pH 8.4 at room temperature for 1 h and separation of the labeled particles from the free dye by gel filtration chromatography using Sephadex G-25 fine resin.

### 2.3. Preparation of Phase-Separated Supported Lipid Bilayers (PSLBs)

For the formation of PSLBs, the following mixtures or pure substances were used: DOPC (18:1), bSM, and Chol at a molar ratio of [2:2:1]; DOPC (18:1) and DPPC (16:0), DOPC (18:1) and DietherPC (16:0), and DOPC (18:1) and DietherPC (18:0). In general, the lipid(s) was/were dissolved in chloroform, mixed in an appropriate glass vial, and the solvent was evaporation using N_2_ gas. Before use, the lipid film was rehydrated using PBS (overall lipid concentration of 10 mg/mL) by vigorous vortexing. After ultra-sonicating at 60 °C for 20 min (the turbidity change confirmed the formation of small unilamellar vesicles), an aliquot was diluted in PBS buffer at a ratio of 1:10 and deposited on a freshly cleaved mica surface; 2 mM of CaCl_2_ was added to the vesicle suspension to improve their spread. The surface was sealed with a home-built chamber and incubated at 55 °C for 10 min. The sample was rinsed at the same temperature at least 10 times with PBS and cooled to 25 °C within 30 min. At this stage, the L_d_ phase marker DiI was incorporated into the formed lipid bilayer by adding it to the solution. The concentration was between 10^−2^ and 10^−3^ mol%. After 20 min, the sample was rinsed again with PBS to remove non-incorporated DiI. After reaching room temperature, the sample was incubated with an HDL particle solution for approx. 2 min and was washed extensively. For ternary membrane systems, such as Chol/bSM/DOPC, the sample was incubated with 2.8 µg/mL HDL. For all other membrane systems, the sample was incubated with 5 µg/mL HDL. All other preparation steps remained identical. To quantitatively compare the association of protein-bound Atto 647 with the L_o_ and L_d_ phases, for DOPC (18:1), bSM, and Chol, we determined the number of single protein-bound Atto 647 signals in the two phases; for all other samples the overall Atto 647 signal was determined. A binary mask obtained from intensity-thresholded DiI signals was used for phase discrimination. The partition coefficient *K_P_* $=I_{1}/I_{2}\cdot A_{2}/A_{1}$ where *I_i_* represents the fluorescence intensity in phase *i* and *A_i_* its area. Statistical differences between samples were assessed using the non-parametric Kruskal–Wallis test (α = 5%). Prior to this, normality was tested using the Shapiro–Wilk test and the homogeneity of variances using Levene’s test. Due to significant results (*p* < 0.05) in the Levene’s test, indicating unequal variances, the Kruskal–Wallis test was deemed appropriate. Post hoc analyses were performed using Dunn’s test with Bonferroni correction. Statistical tests were conducted using JASP (v0.18.3.0).

### 2.4. Preparation of Giant Unilamellar Vesicles (GUVs)

GUVs were generated by electroformation [40] using custom-built GUV Teflon chambers with two platinum electrodes. This preparation technique produces vesicles with varying sizes from 10 to 100 µm. 

For measurements of the diffusion constant and partition coefficient: A volume of 6 µL of lipid (DOPC (18:1) or DOPC (18:1): DietherPC (18:1), molar ratio [3:1]) dissolved in chloroform (overall lipid concentration 1 mg/mL) was homogeneously distributed on the two electrodes, dried with nitrogen, and placed in 300 nM (370 μL) sucrose solution. Electroformation was performed at 2 V and 10 Hz for 1 h, followed by 2 V and 2 Hz for 30 min. After GUV formation, 1 µL DPPE-ASR (10 µg/mL in DMSO) was added to 100 µL GUV solution. 

For measurements of the GP value/hydrogen bond network strength: 20 mL of a 10 mM lipid solution (POPC (16:0–18:1), DietherPC (18:1, 18:0, or 16:0), DPPC (16:0), DOPC (18:1), DOPC (18:1) with varying Chol concentration (molar ratio [9:1], [4:1]) in chloroform containing the fluorescent L_d_ phase lipid marker DiI (0.1 mol%) and/or Laurdan (methanol solution, molar ratio of fluorescent probes to lipids was 1:100) was deposited on Pt electrodes and the solvent was evaporated by a constant N_2_ gas flow for 20 min. In total, 300 µL of 300 mM sucrose was added to a headed (60 °C) and modified Lab-Tec^®^ chamber (Fisher Scientific): two holes with a distance of 5 mm were placed on the chamber’s cap for the electrodes. The electrodes with dried lipids were placed into the sucrose solution and a voltage of 2 V at 10 Hz was applied for 2 h. The temperature was kept at 60 °C. Afterwards, the solution was allowed to cool down to room temperature. After reaching room temperature, the sample was incubated with a 2.8 µg/mL HDL particle solution for approx. 2 min. In total, 300 µL of a 300 mM glucose solution was added to the solution for imaging (i.e., density change causes GUVs to sink). The images were acquired 20 min after the addition of the HDL particle solution by confocal microscopy. Partitioning was measured with confocal microscopy, and membrane order was quantified by Laurdan microscopy/TDFS.

### 2.5. Confocal Microscopy

GUVs were imaged with a laser scanning confocal microscope (LSM 700 AxioObserver, Zeiss, Oberkochen, Germany). The microscope was equipped with a Plan-Apochromat 63x/1.40 Oil DIC M27 objective (Zeiss). The LSM 700 operates with solid-state lasers (polarization-preserving single-mode fibers) at a wavelength of 639 nm and 488 nm. The laser power was adjusted to 0.1–0.5% of the total laser power, equivalent to 2–10 μW. Signals were detected after appropriate filtering on a photomultiplier tube. Typically, a z-stack with a step size of 500 nm was generated. Detector amplification, laser power, and pinhole opening were kept constant for all measurements. 

The images were analyzed by utilizing an algorithm that automatically identifies the interface between the membrane and the inner section of the GUVs. This was carried out by investigating the Atto 647 signal. The information was used to obtain the lateral membrane geometry and yields a binary mask. Fluorescence images (in one image several GUVs were visible) were background-corrected and the GUV surfaces were extracted via a mask. A GUV was selected by defining an ROI of the image. The total amount of fluorescence signal was calculated in both color channels (BodipyFL and Atto 647). Calculating the density was performed by dividing the total amount value by the number of counted masked pixels or by the pure circle length in a pixel. For each experiment, about 50 GUVs were evaluated.

### 2.6. Fluorescence Correlation Spectroscopy (FCS)

Ibidi chambers (#1.5) were coated with BSA (1 mg/mL) and rinsed with PBS after 30 min of incubation. GUVs were allowed to settle in glass-bottomed Ibidi chambers (#1.5) for 15 min. Fluorescence correlation spectroscopy measurements were conducted on the GUVs’ top membrane using an LSM 700 AxioObserver (Zeiss). The microscope was equipped with a Plan-Apochromat 40x/1.2 water objective (Zeiss). The LSM 700 operates with a solid-state laser at a wavelength of 639 nm. The laser power was adjusted to 0.1–0.5% of the total laser power, equivalent to 2–10 μW. Signals were detected after appropriate filtering on a photomultiplier tube. Three measurements (10 s each) were performed on each GUV. At least five vesicles were measured for each sample. Three technical replicas of each sample were measured. Before incubation with HDL particles, the initial mobility of the tracer lipid DPPE-ASR was determined and afterward the same GUV chambers were incubated for (at least) 30 min with 14 µL of 4.3 mg/mL HDL particles (the final concentration of HDL in the chamber ~600 µg/mL, of which approximately 20% would be Chol [41] in HDL particles = 120 µg/mL) and measured again. The obtained auto correlation curves were fitted with a 2D diffusion plus triplet model, with Focus point (version: win 1.13.156) [42]. 

Statistical differences between samples were assessed using the non-parametric Kruskal–Wallis test (α = 5%). Prior to this, normality was tested using the Shapiro–Wilk test and the homogeneity of variances was tested using Levene’s test. Due to significant results (*p* < 0.05) in both tests, indicating non-normal distribution and unequal variances, the Kruskal–Wallis test was deemed appropriate. Post hoc analyses were performed using Dunn’s test with Bonferroni correction. The statistical tests were conducted using JASP (v0.18.3.0).

### 2.7. Single-Molecule Fluorescence Microscopy

The system is based on a Zeiss Axiovert 200 inverted epi-fluorescence microscope equipped with a 100× NA = 1.45 oil-immersion Plan-Apochromat TIRFM objective (Olympus, Vienna, Austria). Samples were illuminated in an objective-type Total Internal Reflection (TIR) configuration via the epi port using 488 nm light from a solid-state laser (Sapphire 200 mW, Coherent, Dieburg, Germany), 647 nm light from a Kr^+^-laser (Innova 301, Coherent), or 532 nm light from a solid-state laser (Millennia X, Spectra Physics, Vienna, Austria), with intensities of 3–10 kW/cm^2^. After appropriate filtering, the emitted signals were imaged on a back-illuminated, TE-cooled CCD camera (Andor iXon Du-897 BV, Oxford Instruments, Belfast, UK). For precise control of the illumination timings, acousto-optical modulators (1205C, Isomet, Gröbenzell, Germany) were used. Timing protocols were generated by an in-house program package implemented in LABVIEW (National Instruments, Vienna, Austria). Illumination times were between 1 and 5 ms. Image series were recorded with a delay between two consecutive images of 15 to 300 ms.

### 2.8. Laurdan Time-Dependent Fluorescence Shift (TDFS)

The temperature in the cuvette holders was maintained using a water-circulating bath at 23 ± 0.5 °C. Steady-state excitation and emission spectra were acquired using a Fluorolog-3 spectrofluorometer (model FL3-11; Jobin Yvon Inc., Edison, NJ, USA) equipped with a xenon arc lamp. The steady-state spectra were recorded in steps of 1 nm (bandwidths of 1.2 nm were chosen for both the excitation and emission monochromators) in triplicate and averaged. Fluorescence decays were recorded on a 5000 U single-photon counting setup using a NanoLED 11 laser diode (375 nm peak wavelength, 1 MHz repetition rate) and a cooled Hamamatsu R3809U-50 microchannel plate photomultiplier (IBH, Glasgow, UK). A 399 nm cut-off filter was used to eliminate scattered light. The signal was kept below 1% of the repetition rate of the light source. Fluorescence emission decays were recorded at a series of wavelengths spanning the steady-state emission spectrum (400–550 nm) in steps of 10 nm. Data were collected until the peak value reached 5000 counts. The full width at half maximum (FWHM) of the instrument response function was 78 ps.

### 2.9. Laurdan GP

A steady-state emission spectrum (*Ex* = 378 nm) and two excitation spectra (Em = 440 and 490 nm) were recorded. The excitation spectra were used to calculate excitation generalized polarization spectra (*GP_EX_*) [43]:$${GP}_{EX}\left(\lambda_{EX}\right)=\frac{I_{440}-I_{490}}{I_{440}+I_{490}}$$
where *I*_440_ and *I*_490_ represent the fluorescence intensity emitted at 440 and 490 nm, respectively, at the excitation wavelength *λ_EX_*.

### 2.10. Laurdan TDFS Analysis

The fluorescence decays were fitted to a multi-exponential function via the reconvolution method using IBH DAS6 software (https://www.horiba.com/int/scientific/products/detail/action/show/Product/das6-1377/, accessed on 3 December 2024). The purpose of the fit is to deconvolve the instrumental response from the data and should not be over-parameterized. The fitted decays, together with the steady-state emission spectrum, were used for the reconstruction of time-resolved emission spectra (TRES) by a spectral reconstruction method [44]. The reconstruction routine was implemented in MATLAB 24.2. The position of TRES’ maximum *ν*(*t*) and its FWHM(t) were inspected. The two main parameters describing the polarity and mobility of the probed system were derived from *ν*(*t*). The total amount of fluorescence shift Δ*ν* reflects the polarity of the environment of the probe and is calculated as: Δ*ν* = *ν*(*0*) − *ν*(*∞*) where ν(0) = 23,800 cm^−1^ is the position of TRES maximum at *t* = 0, estimated using the method of *Fee and Maroncelli* [45] and *ν*(*∞*) is the position of the TRES at the fully relaxed state. The TDFS kinetics depend on the dynamics of the polar moieties in the vicinity of the probe and can be expressed as the integrated relaxation time *τ_r_*:$$\tau_{r}=\int_{0}^{\infty }\frac{\nu \left(t\right)-\nu \left(\infty \right)}{\Delta \nu }dt$$

The intrinsic uncertainties for the TDFS parameters were 50 cm^−1^ and 0.05 ns for Δ*ν* and *τ_r_*, respectively.

### 2.11. Single Molecule/Particle Tracking

Individual diffraction-limited fluorescence signal spots were selected, fitted with a Gaussian intensity profile, and tracked using in-house algorithms implemented in MATLAB (MathWorks); the single-molecule positions were obtained with an accuracy of $\sigma_{xy}=20-40\;\mathrm{nm}$. Diffusion constants were determined as described previously [46]. In brief, trajectories are characterized by a sequence of positions $\vec{x\;}(i)$, with *i* ranging from 1 to the number of observations of this signal. The mean square displacement $\langle r^{2}\rangle$ was calculated as a function of the time lag $t_{lag}=n(t_{ill}+t_{delay})$ according to $\langle r^{2}\rangle ={\langle {\left(\vec{x}\left(i\right)-\vec{x}\left(n+i\right)\right)}^{2}\rangle }_{i=1;1+n,1+2n;\ldots }$, with n denoting the difference in frame index *i*. Data were analyzed by fitting with $\langle r^{2}\rangle =4Dt_{lag}+4\sigma_{xy}^{2}\;(Equation\;(1))$, yielding the lateral diffusion constant D and the single-molecule localization precision $\sigma_{xy}$.

To discriminate between two mobile fractions (see Appendix A), a bimodal model as previously described was used [46]. Briefly, the cumulative density function of the square displacements (sd) was fitted with $cdf=1-\alpha \cdot exp\left[-\frac{sd}{{msd}_{1}\left(t_{lag}\right)}\right]-\left(1-\alpha \right)\cdot exp\left[-\frac{sd}{{msd}_{2}\left(t_{lag}\right)}\right]\;(Equation\;(2))$, yielding the proportion of the two mobile components, α and (1 – α), and the t_lag_-dependent mean square displacements *msd*_1_ and *msd*_2_. Confined diffusion (diffusion constant *D*) within an impermeable circle of radius *R* was fitted with $msd=R^{2}\left(1-exp\left[-\frac{4Dt_{lag}}{R^{2}}\right]\right).$

## 3. Results

Phosphatidylcholine (PC) lipids play an important role in interacting with HDL particles, given their role as the major outer leaflet lipid of mammalian cell membranes [40]. This work studies the interaction of HDL particles with membranes made of different PC lipids, whose structure differs in the linkage (i.e., ester- vs. ether-linked) of the glycerol region at sn-1 and sn-2 to, and the length/saturation of two acyl chains (see Figure 1A and Appendix A). A polar headgroup was attached at glycerol’s sn-3 via a phosphodiester bond. Planar membranes were primarily used to determine phase preferences, while spherical membranes were used to analyze diffusion. In both cases, the interaction with HDL particles was examined.

### 3.1. HDL Particles Transfer Cargo Molecules via L_d_ Phase Interaction

It is speculated that the plasma membrane is divided into relatively L_o_ and L_d_ lipid environments on nanometer length scales, where L_o_ phases represent the hypothesized “lipid rafts”, mimicking low-fluidity membranes [47]. Membrane fluidity is a dynamic parameter that varies depending on the cell type and state; therefore, it is important to understand how it affects HDL particle interaction. This speculation finds support through domain formation occurring in synthetic lipid vesicles and bilayers containing saturated glycerophospholipids or sphingolipids and Chol [48]. Such biomimetic bilayers are frequently utilized as minimal systems that maintain biological relevance such as the ternary model membrane system composed of DOPC (18:1), Chol, and brain Sphingomyelin (bSM) [1]. Experiments conducted on phase-separated supported lipid bilayers (PSLBs) using a 2:2:1 mixture of DOPC (18:1), bSM, and Chol yielded two distinct phases (Figure 1B). The L_d_ phase is mostly enriched in DOPC (18:1), while the L_o_ phase is solely composed of bSM and Chol [7]. To visually distinguish the phases, specific markers such as the fluorescent lipid DiI can be used, which predominantly populate the L_d_ phase [34] (Figure 1B, left image). 

After the addition of HDL particles, the fluorescently labeled HDL-associated proteins (HDLap) yielded a clear preference for the Ld phase (Figure 1B, middle image). The calculated partition coefficient for HDLap is *K_p_* = 3 a.u. ± 1.2 a.u. (standard deviation) for phase 1 (L_d_) and phase 2 (L_o_), as illustrated in Figure 1B (right-hand image). Approximately 70% of all HDLap fluorescence signals were immobile; the remaining 30% showed confined diffusion (Appendix A; Movie S1). Drawing from previous studies [37], these diffusion data suggest that most HDL particles were immobilized due to fusion, while the remaining fraction is primarily attached to the membrane and thus shows confined diffusion. A predominant localization of HDLap was detected at the phase interfaces (~30% of all signals), indicating interaction with phase boundaries. Line tension at phase boundaries may offer anchor points for fusion processes through the energy generated by the hydrophobic mismatch between the L_o_ and L_d_ phases being comparable to that required to form a lipid stalk intermediate [49]. The Chol-BodipyFL transferred by the HDL particle into the L_d_ shows free diffusion (Appendix A). Notably, after maintaining contact with the HDL particles, Chol-BodipyFL is transferred into the L_d_ phase and triggers a rearrangement within the L_o_ phase, while the HDLap signal is retained in its initial environment (Appendix A). In summary, these results suggest that HDL particles interact with fluid membranes predominately to facilitate cargo exchange. The next step will be to determine which specific biophysical properties of the target membrane, particularly the lipids, are responsible for enabling protein interaction or inducing cargo transfer.

### 3.2. Phospholipid Backbone Influences HDL Interaction: Ester vs. Ether

We compared two lipids with identical fatty acid chains, but where the ester linkage is replaced by an ether one, in experiments with giant unilamellar vesicle (GUV) membranes using a pure DOPC (18:1) or a [3:1] molar mixture of DOPC (18:1) and DietherPC (18:1). The membranes composed of pure DOPC (18:1) or a mixture of DOPC (18:1) and DietherPC (18:1) did not exhibit distinguishable phases; instead, they formed a homogeneous L_d_ phase membrane. The overall effect of HDL particle interaction and the associated cargo transfer can be determined indirectly using an indicator lipid (i.e., DPPE-ASR). DPPE-ASR is used as an L_d_-phase [9] marker, though the lipid can likewise be used to indirectly determine overall changes in the membrane due to an altered diffusion behavior [50]. Noteworthily, no difference regarding the lateral diffusion constant of the indicator lipid was observed before HDL interaction (Figure 2A).

However, adding HDL particles yielded statistically significant differences: while in DOPC (18:1) membranes the indicator lipid’s lateral diffusion constant decreased by 15%, in DOPC (18:1)/DietherPC (18:1) [3:1] lipid mixtures it decreased by 20%. In general, the decrease in diffusion constant can be attributed to HDLap anchoring in the membrane and to the alteration of the target membrane’s composition through cargo transfer (e.g., the increased Chol content of the target membrane) [37]. The more pronounced decline in the DOPC (18:1)/DietherPC (18:1) [3:1] lipid mixture relative to pure DOPC (18:1) may be attributed to an augmented interaction with HDL, encompassing binding and cargo transfer. An alternative explanation could be that the quantitatively similar interaction has a greater effect on the DietherPC (18:1) than on DOPC (18:1) lipids.

### 3.3. HDL Particles Strongly Interact with Saturated Ether Lipid Phases

The impact of the chain length and glycerol linkage between the fatty acid chain and the head group on HDL particle interaction was investigated using planar membranes composed of two-component mixtures. Their two-dimensional nature facilitates the analysis of lipoprotein particle partitioning. When DOPC (18:1) is mixed with DPPC (16:0), a two-phase system is formed consisting of a DOPC (18:1) L_d_ phase and a DPPC (16:0) gel phase. After the addition of HDL particles, the calculated partition coefficient of the HDLap fluorescence signals was *K_p_* = ${68\cdot 10}^{-3}\pm {22\cdot 10}^{-3}$ a.u. (Figure 2B, Appendix A, first row). In comparison to our measurement of a DOPC/bSM/Chol mixture (Figure 1B), wherein HDLap predominantly interacted with the L_d_ phase, the partition coefficient is nearly two orders of magnitude lower for the DOPC/DPPC mixture. This shows that the disparity in HDLap signal density (i.e., *K_p_* value) between the two phases is more pronounced for the DOPC/DPPC PSLB than previously observed for the DOPC/bSM/Chol PSLB. This may be due to the decreased area per lipid of DPPC [51,52] and thus the stronger hydrogen bond network within the gel phase. Diether lipids with different chain lengths were compared to evaluate any influences of the fatty acid chain length. Similarly to DPPC (16:0) [53], saturated DietherPC lipids form gel phases. In general, a higher disparity in HDLap signal density was observed for both DietherPC lipids compared to DPPC (16:0) (Appendix A, middle and bottom illustration, *K_p_* (DietherPC (18:0)) = ${40\cdot 10}^{-3}\pm {3\cdot 10}^{-3}$ a.u., *K_p_* (DietherPC (16:0)) = ${6\cdot 10}^{-3}\pm {3\cdot 10}^{-4}$ a.u. Figure 2B). Additionally, the observation from previous measurement (Figure 2A) is supported, as the linkage of the glycerol region influences HDL interaction. However, an equally decisive factor is the length of the fatty acid chains. A comparison of the partition coefficient of DietherPC (18:0) with DietherPC (16:0) reveals a significant difference (Figure 2B). In summary, enhanced interactions are observed in two-phase lipid mixtures when ether linkages (DietherPC (18/16:0)) are present compared to ester linkages (DOPC (18:1)). Additionally, these interactions are further amplified by shorter fatty acid chain lengths (DietherPC (18:0) vs. DietherPC (16:0)).

### 3.4. A Multi-Correlative Analysis of Individual Lipid Properties Reveals Differences in HDL Particle Interactions and Cargo Transfer

To attain a more precise understanding of how the glycerol region mobility and thus its hydrogen network influences the interaction and subsequent cargo (i.e., Chol-BodipyFL) transfer of HDL particles, lipid vesicles consisting of DOPC (18:1), DOPC (18:1) with varying Chol content (10% and 20% Chol), POPC (16:0–18:1), DPPC (16:0), and DietherPC (16:0) were investigated. The cholesterol concentrations in DOPC membranes were not increased further, as this resulted in relatively unstable and non-reproducible target membranes, most likely caused by the incipient asymmetry of the lipid bilayers [54]. The integration of HDL-associated components (Chol-BodipyFL and HDLap-Atto 647) and the localization of the HDL particles within GUVs, produced from different lipids or lipid mixtures, (Appendix A) was analyzed with confocal microscopy and correlated with Laurdan’s GP value and TDFS experiments [55], both yielding information on glycerol region mobility [56]. The GP value of Laurdan is an empirical steady-state ratiometric parameter that reports on both the hydrogen bond network and the mobility of carbonyl groups in the membrane. Nonetheless, the GP value can be taken as an indicator of glycerol region mobility but not as an indicator of the extent of the hydrogen bond network, “or water penetration”, of a fully hydrated, fluid L_d_ lipid bilayer, as often found in the literature [55]. Laurdan, which is located near the DOPC (18:1) sn-1 carbonyl group [57], makes it possible to sense the mechano-elastic properties of the linkage region of the lipid bilayer. With increasing GP value and thus decreasing glycerol region mobility, we found for L_o_ and L_d_ phase lipids a significant decrease in the average HDLap-Atto 647 and Chol-BodipyFL signals. 

To quantify the initial membrane environment’s influence on HDL particle interaction, we performed TDFS experiments [55], which measure the spectral shift during the relaxation of a fluorophore caused by its local environment (Figure 3A). Analysis of the frequency shift Δ*ν* and the relaxation time *τ_r_* yields independent information on the probe’s environment polarity and mobility, i.e., the level and strength of the hydrogen bond network in its vicinity, respectively. Although the changes in Δν are close to the error, the measured data predict an increase in the hydrogen bond network for the lipid bilayers in the L_d_ phase. Interestingly, the increase in Δ*ν* by the addition of Chol indicates increased probe polarity although the variations are very small and close to the error. This also confirms that interpreting GP values as changes in the level of the hydrogen bond network is incorrect [55]. With increasing GP value and thus decreasing glycerol region mobility, we found for L_o_- and L_d_-phase lipids a marked decrease in the average HDLap-Atto 647 and Chol-BodipyFL signals (Figure 3B). In contrast to this trend, HDL particle interaction was highest for the DietherPC (16:0) gel phase, whereas the overall lowest value was detectable with DPPC (16:0). The estimated relaxation times for the DietherPC (16:0) and DPPC (16:0) lipids evidence their significantly slower relaxation, i.e., restricted glycerol region mobility, experienced in these gel-phase samples in comparison to all others in the L_d_ phase. GP correlates with the magnitude of *τ*_r_. Therefore, a higher GP and higher magnitudes of *τ*_r_ signify a slower relaxation process, i.e., the reduced mobility of the sn-1 carbonyls of the lipids in the L_d_ phase. This can be related to an increase in the strength of the hydrogen bond network at the level of the sn-1 carbonyls in the L_d_ phase. 

However, the value for the estimated maxima for the DietherPC (16:0) is about half of the value determined for the DPPC (16:0) bilayer. Apparently, even being in the gel phase the glycerol region of the ether lipids showed a remarkably high mobility. Ether lipids lack the carbonyl groups that strongly contribute to the hydrogen bond network in the glycerol region of lipids. Thus, the hydrogen bond network in the glycerol region between the alkyl chain and the headgroup is an essential factor for mediating HDL particle interaction with lipid bilayers. However, when comparing protein interaction to cargo transfer, it becomes clear that the ratio reverses for gel-phase lipids, regardless of the glycerol binding site. This indicates that cargo transfer is favored by increased fluidity and reduced mobility in the glycerol region.

## 4. Discussion/Conclusions

Lipid composition is an important mediator of membrane fusion as it alters the physical properties of the membrane as well as the structure, organization, and dynamics of the fusion proteins [58]. In fact, some lipids bind directly to proteins and control their behavior; for example, by affecting membrane curvature [59]. Additionally, the membrane interface, particularly the dynamics of the hydrogen bond network, plays a decisive role in protein interactions [24,60]. This study focuses on the receptor-independent interaction of HDL particles and plane PC lipid membranes. The impact of specific properties of the fatty acid chains, including their length and degree of saturation, and of the glycerol linkage (ester vs. ether) was evaluated. In conjunction with ternary membrane mixtures, the relationship between HDL interaction and glycerol region mobility, and the hydrogen bond network, and consequently lipid membrane polarity, was subjected to a comprehensive investigation. Three-component mixtures demonstrate that HDL particles predominantly interact and transfer their cargo in the fluid phase, primarily composed of DOPC (18:1). Lipids with identical fatty acid chain lengths and degrees of saturation, but with an ester (i.e., DOPC (18:1)) instead of an ether (i.e., DietherPC (18:1)) glycerol linkage, exhibit a diminished interaction with HDL particles. This indicates that lipids with a stronger hydrogen bond network interact preferentially with HDL particles. In this particular case, it might be that ether lipids within a DOPC membrane facilitate the initial contact and interaction with the HDL particle and subsequently facilitate cargo exchange within the DOPC membrane. In general, saturated gel-phase lipids (i.e., DPPC (16:0)) exhibit even stronger hydrogen bond network characteristics. However, there were significant differences in the interaction between ester- and ether-linked lipids, with the ether bonds leading to a stronger interaction due to decreased glycerol region mobility. Ether lipids lack the carbonyl groups that strongly contribute to the hydrogen bond network in the glycerol region of lipids. Thus, the hydrogen bond network in the glycerol region between the alkyl chain and the headgroup is an essential factor for mediating HDL particle interactions with membranes. Both gel-phase and fluid-phase membranes show interactions with HDL particles. However, there are significant differences in the nature of these interactions. While binding occurs preferentially in gel-phase membranes, fluid-phase membranes facilitate cargo transfer. Cholesterol in the target membrane reduces the interaction with HDL particles by increasing the packing density and stiffness of the membrane, thereby modulating its fluidity. Cholesterol is small enough to fill the space between lipid head groups and thus influences the overall stiffness and fluidity of membrane components [61]. Recently, a study conducted by Yesylevskyy et al. [62] showed by atomistic simulation that the order parameter of lipid tails decreases significantly in curved membranes, whereas the area per lipid increases in the convex monolayer and decreases in the concave monolayer. The concomitant lipid composition based on membrane curvature massively influences the interaction. In addition to the cholesterol redistribution within the membrane, curvature modulates phase separation in lipid bilayer membranes [63]. In addition, lipid properties such as saturation state, headgroup size, and chain length, play an essential role in protein interaction [64]. A less noticed property of lipids is their hydrophilicity/hydrophilicity within the membrane assembly [65]. Moreover, increased Chol content reduces the lipid’s hydrogen bond network [66], reduces passive permeability, and increases the GP value [28]. In addition to the hydrophobic and van der Waals forces, the 3-OH group of Chol to PC lipids forms H-bonds with sn-2 carbonyl and phosphate groups in the glycerol linkage region of the bilayer. Membrane fluidity and hydrogen bond network strength change between different membrane phases [67]. Moreover, the length of the fatty acid chain also appears to influence the interaction, likely due to a decrease in membrane rigidity and thus increased mobility [68]. Lipid transfer from lipoprotein particles to membranes depends on the properties of the lipid itself and the contact process with the plasma membrane. Moreover, membrane curvature affects lipid exposure and, consequently, the contact process. In future studies, we aim to focus more closely on the influence of membrane curvature in conjunction with the corresponding lipid composition. Consequently, the plasma membrane itself yields a high binding capacity for HDL particles.

In summary, the influence of chain length and the degree of saturation of the fatty acid chains, as well as the cholesterol content in the membrane, could be discerned. Major factors for increased interaction between lipid membranes and HDL particles are properties such as glycerol region mobility. By increasing the accessible surface area in the glycerol region, interaction occurs preferentially. However, the data indicate that neither the hydrogen bond network nor the glycerol region mobility alone are sufficient to fully explain the interaction with HDL particles.

## Figures and Tables

**Figure 1 membranes-14-00261-f001:**
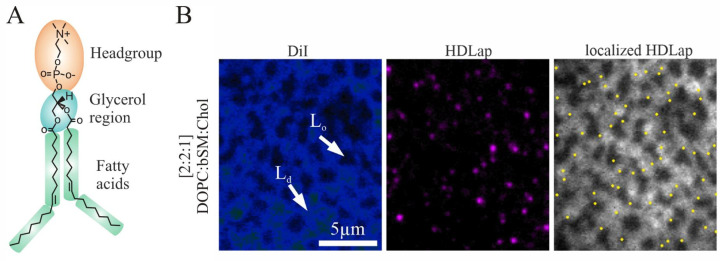
(**A**) The nomenclature used, exemplified by a structural model of a single DOPC (18:1) lipid. (**B**) HDL-associated protein (HDLap, fluorescently labeled with Atto 647, purple) partitions preferential to the L_d_ phase (blue) in PSLBs ([2:2:1] mixture of DOPC (18:1), bSM, and Chol) supported on a glass surface. The bilayer was treated with DiI as an L_d_ phase marker (shown in the first color channel, corresponding to the image on the left) and HDL particles (shown in the second color channel, corresponding to the middle image, and represented as yellow dots in the right image, illustrating the center of mass.) (partition coefficient of *K_p_* = 3 a.u. of HDLap (phase 1: L_d_, phase 2: L_o_)). The last image displays the positions of single HDLap-Atto 647 signals (indicated as yellow dots) overlaid with the DiI fluorescence image.

**Figure 2 membranes-14-00261-f002:**
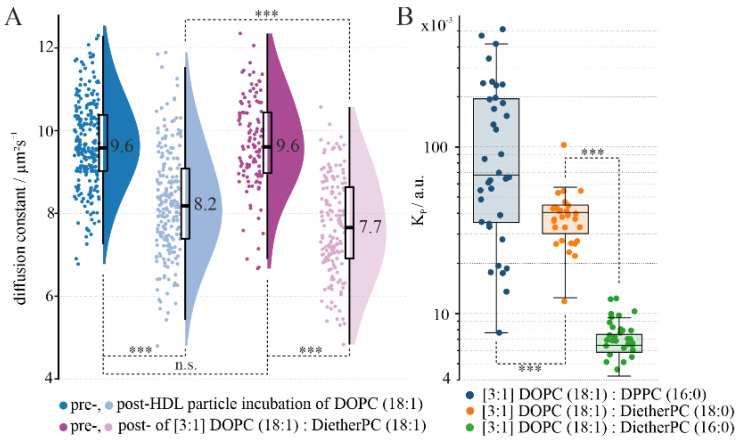
An HDL particle interaction with different lipid membranes. (**A**) shows the scatter, box and distribution plots of the lateral diffusion constant of DPPE-ASR within a GUV membrane of DOPC (18:1) and DOPC (18:1): DietherPC (18:1) [3:1] pre and post incubation with an HDL particle solution. Each individual point depicts the diffusion constant obtained from one measurement on the top GUV membrane: pre HDL incubation (dark blue) (DOPC (18:1) preHDL = 9.6 μm^2^/s ± 0.1 μm^2^/s, N = 237) and (dark magenta) (DOPC (18:1): DietherPC (18:1) preHDL = 9.6 μm^2^/s ± 0.1 μm^2^/s, N = 114), post HDL incubation (light blue) (DOPC (18:1) postHDL = 8.2 μm^2^/s ± 0.1 μm^2^/s, N = 208) and (light magenta) (DOPC (18:1): DietherPC (18:1) postHDL = 7.7 μm^2^/s ± 0.1 μm^2^/s, N = 153) (n.s. not significant, *** *p* < 0.001@ α = 5%). (**B**) shows the boxplot chart of the calculated partitioning coefficient *K_p_* (phase 1: L_d_ phase (i.e., DOPC); phase 2: gel phase (i.e., DPPC, DietherPC)) regarding HDLap localization in respect of three different (i.e., lipid composition) PSLBs. (Median ± SE) DOPC (18:1): DPPC (16:0) [3:1] (blue): *K_p_* = 68 · 10^−3^ ± 22 · 10^−3^ a.u. (N = 36); DOPC (18:1): DietherPC (18:0) [3:1] (orange): *K_p_* = 40 · 10^−3^ ± 3 · 10^−3^ a.u. (N= 30); DOPC (18:1): DietherPC (16:0) [3:1] (green): *K_p_* = 6 · 10^−3^ ± 3 · 10^−4^ a.u. (N = 30) (*** *p* < 0.001 @ α = 5%).

**Figure 3 membranes-14-00261-f003:**
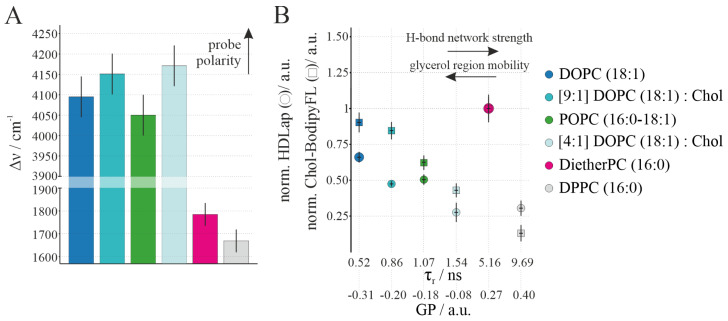
An HDL particle interaction with lipid membranes of different probe polarity/GP value/hydrogen bond network strength. (**A**) The bar chart depicts the total amount of fluorescence shift Δ*ν* of the Laurdan emission for different GUV membrane compositions. The intrinsic uncertainty for the Δ*ν* parameter is 50 cm^−1^. The Δν parameter is directly related to the hydrogen bond network level of sn-1 carbonyls. A higher magnitude of fluorescence shift reflects a higher probe polarity. (**B**) shows the relaxation time *τ_r_*/GP value for different GUV membranes (Supplemental Appendix A) in relation to normalized (i.e., both DietherPC (16:0) values were set to 1) HDL-associated protein (HDLap) Atto 647 (○) and the normalized Chol-BodipyFL (□) signal per pixel (Supplemental Appendix A). A higher magnitude of GP value and relaxation time *τ_r_* reflect a lower glycerol region mobility and stronger hydrogen bond network, respectively. Error bars indicate the standard error of the mean.

## Data Availability

The data presented in this study are available from the corresponding author upon request. The data are not publicly available due to restrictions of the involved scientists and facilities.

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
