# Peer review of "“Head-to-Toe” Lipid Properties Govern the Binding and Cargo Transfer of High-Density Lipoprotein"

_membranes, 2024, doi:10.3390/membranes14120261_

Round 1

Reviewer 1 Report

Comments and Suggestions for Authors

The authors performed several experiments, based on the recording of fluorescent signals, with accurate data analysis aimed at a better understanding of the interaction of HDL particles (opportunely labelled) with target membranes. They accurately design different monophasic and phase-separated in vitro lipid model systems (in the form of bilayers and GUVs) to evaluate the effect of cholesterol, acyl chain length and hydrogen bonding in the hydrophilic headgroups on HDL particles-membrane interaction. They also tend to disentangle the effect of binding/interaction of HDL particles with membranes and cargo (cholesterol) release.

Although their results show that the HDL particles-membrane interaction clearly depends on acyl chain length as well as on the nature of hydrophilic headgroup (ester vs ether), which is strictly related to hydrogen bond network, the reported experiments are not sufficient to dissect the influence of HDL particles binding (to membranes) onto cargo (cholesterol) release/transfer.

·         along the text, references are missing for some of the author’s statements. For example, “It is speculated that the plasma membrane is divided into relatively Lo and Ld lipid environments on nanometre length scales, where Lo phases represent the hypothesized "lipid rafts", mimicking low fluidity membranes” in paragraph 3.1; “The cholesterol concentrations in DOPC membranes were not increased further, as this resulted in relatively unstable and non-reproducible target membranes.” in paragraph 3.4, and the following sentences in the introduction “Ether lipids are a substantial structural component in cell membranes, representing around 20% of the total phospholipid content in mammals.”; “These interfaces are crucial for important cellular processes such as protein binding or drug interaction”.

Introduction

·         “Chol is an essential building block with up to 40 mol% [13] in eukaryotic plasma membranes and regulates the mem-brane’s fluidity.”. Following the same reference ([13]: van Meer, Gerrit; Voelker, R. Dennis; Feigenson, W.G. Membrane Lipids : Where They Are and How They Behave. Nat 587 Rev Mol Cell Biol. 2009, 9, 112–124) as well as Raghava S, Giorda KM, Romano FB, Heuck AP, Hebert DN. The SV40 late protein VP4 is a viroporin that forms pores to disrupt membranes for viral release. PLoS Pathog. 2011;7(6). doi:10.1371/journal.ppat.1002116, the cholesterol content can go up to 50%.

·         This phase (Lo) shares similarities with the Ld phase in respect to a reduced lateral lipid motion.”. I find this sentence unclear. Lipids partitioning in the Lo present a reduced motion with respect to lipids in Ld. Hence it is not a similarity in between the two phases. Also a reference should be introduced.

Materials and Methods

·         Nomenclature of HEPES to be checked.

·         Definition of dialysis buffer (what do the percentages of NaCl and EDTA means?).

·         Minor comment: The laser power employed for the experiments is not reported.

·         In section 2.3 the authors explain how they prepare Phase-Separated Supported Lipid Bilayers samples. They employ two different concentrations of HDL without commenting on the need of more concentrated incubation steps. Also it is not clear for which samples they increase HDL concentration (they wrote “all other samples”).

·         Minor comment: I suggest to report the wavelength of Ex and Em of all the dies.

·         The authors refer to Fig. S3 when talking about two mobile populations and a bimodal model used to interpret the data (and obtain the population fractions). However, such Figure does not show HDL particles localization, neither refers (in the caption) to two mobile fractions. Maybe the authors could include another supplementary movie recorded on one the PSLB shown in Fig S3. Moreover, they do not comment on the reasons for the presence of two populations.

Results

·         In paragraph 3.1 the authors claim “the fluorescently labelled HDL-associated pro-teins (HDLap) yielded a clear preference for the Ld phase (Figure 1B, middle image).” referring to the middle inset in Fig 1B, however, such preference would be much more evident if they superimposed the images (as they did for the right inset). Therefore, to enhance comprehension, Fig 1 B could be reduced to 2 images showing the phase separation, and the superposition (as they did in Fig S3). Moreover, regarding the localization and the right panel in Fig.1B, I suggest the authors to briefly explain how they perform such localization: are the yellow spots the center of mass of the HDL particles with relative error position? Or do they represent the real size of the HDL particles?

·         Minor comment: to improve the comprehension, I suggest to modify Fig 1 A and add in the same panel an example of diether lipid.

·         The authors claim that “…these results suggest that HDL particles interact with fluid membranes predominately to facilitate cargo exchange”. They also assess in line 475 that “when comparing protein interaction to cargo transfer, it becomes clear that the ratio reverses for gel-phase lipids, regardless of the glycerol binding site”. However, the preferential binding that they observe to Ld phase can be due to a better anchoring of the HDL particles to the DOPC lipids (rather than SM and cholesterol), hence leading to increased cholesterol release. To distinguish between the two mechanism (binding and cargo transfer), other experiments would be required, for example employing empty (cholesterol-free) HDL particles, to rule out the effect of cargo transfer. Although they show (Fig S2C) that cholesterol was mainly released in Ld phase, that experiment was performed with the ternary mixture in which HDL particles partition within the Ld.  In the following experiments, when HDL particles were bound to the gel phase, they still observe the release of cholesterol. Hence this may suggest that the release is only depending on the anchoring of the HDL and not on the phase of the target membrane. The authors should comment more on that.

·         To enhance result comprehension, the authors should show the distribution in time of chol-BodipyFL in the binary mixtures shown in FigS3, where the HDL particles partition within the gel phase (DOPC:DPPC [3:1], DOPC:Diether PC (18:0) [3:1] and DOPC:Diether PC (16:0)). These data may be useful to gather some more pieces of information regarding cholesterol transfer and partition.

·         The associated error of the Kp in the ternary mixture bilayer is missing.

·         Minor comment: To improve comprehension, the results regarding the partition (Kp) of HDL particles in the ternary DOPC:chol:bSM membranes should be added in Fig 2B.

·         Minor comment: P in Kp is not always subscript.

·         The authors claim “In summary, in a binary mixture with DOPC (18:1) ester instead of ether linkage (DPPC vs. both DietherPC lipids) as well as longer fatty acid chains (Diether PC (18:0) vs. Diether PC (16:0)) yields an enhanced interaction with HDL particles.”, however they show the opposite (HDL particles interact more with dieterPC (Fig 2A) and prefer partitioning in gel phase domains (Fig S3)). Indeed, in the conclusion (line 505) the authors comment that “lipids […] with an ester (i.e., DOPC (18:1)) instead of an ether (i.e., DietherPC (18:1)) glycerol linkage, exhibit a diminished interaction with HDL particles.”.

Other comments

·         Caption of Fig S4 is confusing: “Circles indicate Ld phase membranes, squares indicate gel phase membranes”. But from the axis labels of the graph it is clear that circles are related to HDLap signal and square to chol.

·         Discussion/Conclusion: line 534 “By increasing the accessible surface area in the glycerol region, interaction occurs to occur preferentially” is it a typo?

·         Discussion/Conclusion: line 535 “However, the data indicate that neither the hydrogen-bond network nor the glycerol region mobility alone are suffi-cient to fully explain the interaction with HDL particles.”. Do the authors have perspectives in mind or propose future experiments to better understand HDL-target membrane interaction?

Reviewer 2 Report

Comments and Suggestions for Authors

This is an important study on how different lipid types influence HDL interactions with the membranes. The manuscript is very well written, although sometimes difficult to follow. I believe if the result subsection titles summarized the specific result, it would improve the reading experience.

I also have a few specific comments:

Page 2, line 46: "This phase shares similarities with the Ld phase in respect to a reduced lateral lipid motion. " 
Clarify because previously reduced/low lateral lipid motion was associated with solid-like gel phase. Page 1 line 39: "At sufficiently low temperatures, the solid-like gel [2] phase is characterized by low lateral lipid motion [3], high lipid packing/membrane order and a low level of membrane hydration [4].

page8, line 366:   "This membrane did not show ... " I
Clarify which membrane.

Round 2

Reviewer 1 Report

Comments and Suggestions for Authors

The authors appropriately address the requested modifications in the text, or explain better in their reply. I only have few comments (the comments from the first report are presented in blue, and the authors replies in red. In black you’ll find the new comments).

MINOR COMMENTS

· “Chol is an essential building block with up to 40 mol% [13] in eukaryotic plasma membranes and regulates the mem-brane’s fluidity.”. Following the same reference ([13]: van Meer, Gerrit; Voelker, R. Dennis; Feigenson, W.G. Membrane Lipids : Where They Are and How They Behave. Nat 587 Rev Mol Cell Biol. 2009, 9, 112–124) as well as Raghava S, Giorda KM, Romano FB, Heuck AP, Hebert DN. The SV40 late protein VP4 is a viroporin that forms pores to disrupt membranes for viral release. PLoS Pathog. 2011;7(6). doi:10.1371/journal.ppat.1002116, the cholesterol content can go up to 50%.

Response by the authors: We thank the reviewers for their thorough evaluation of the citations. The lipid composition, particularly in the case of cholesterol, varies not only between mammalian and yeast cells but also among the different membrane systems within a cell. In eukaryotic cells, specifically in the plasma membrane, cholesterol levels can reach up to 40 mol%, as indicated in the cited literature. We have not found any evidence in the references that contradicts this.

Cholesterol levels can go up to 50% (Raghava S, Giorda KM, Romano FB, Heuck AP, Hebert DN. The SV40 late protein VP4 is a viroporin that forms pores to disrupt membranes for viral release. PLoS Pathog. 2011;7(6).

· In paragraph 3.1 the authors claim “the fluorescently labelled HDL-associated pro-teins (HDLap) yielded a clear preference for the Ld phase (Figure 1B, middle image).” referring to the middle inset in Fig 1B, however, such preference would be much more evident if they superimposed the images (as they did for the right inset). Therefore, to enhance comprehension, Fig 1 B could be reduced to 2 images showing the phase separation, and the superposition (as they did in Fig S3). Moreover, regarding the localization and the right panel in Fig.1B, I suggest the authors to briefly explain how they perform such localization: are the yellow spots the center of mass of the HDL particles with relative error position? Or do they represent the real size of the HDL particles?

Response by the authors: In this experiment, a high-resolution single-molecule fluorescence microscope was employed, enabling the visualization of individual proteins as well as lipids. Each HDL lipoprotein particle is estimated to carry approximately 30 cholesterol molecules, resulting in an amplification of the cholesterol-BODIPY signal by a factor of 30 (see Supplementary Figure S2 C).

In Figure 1B (left), the fluorescent lipid DiI was used to visualize the phases, not to detect individual lipids. This image represents the signal from DiI lipids incorporated into the Ld phase, not a superposition of signals. By contrast, Figure 1B (center) displays individual HDLap signals, where each “color dot” typically corresponds to a single HDL particle, unless the particle or protein carries two labels. Since these are single-molecule signals, superposition is not possible.

Figure 1B (right) presents the overlay of the first two images, with the yellow dots indicating the center of mass of the HDLap signals, as the reviewer correctly observed.

I suggest the authors to specify (in the caption) that the yellow spots represent HDL particles center of mass. Moreover, when they write “the fluorescently labelled HDL-associated pro-teins (HDLap) yielded a clear preference for the Ld phase (Figure 1B, middle image).” I suggest them to refer to the right inset to enhance clarity.

· Discussion/Conclusion: line 535 “However, the data indicate that neither the hydrogen-bond network nor the glycerol region mobility alone are suffi-cient to fully explain the interaction with HDL particles.”. Do the authors have perspectives in mind or propose future experiments to better understand HDL-target membrane interaction?

Response by the authors: We sincerely thank the reviewer for this exceptionally intriguing question! Yes, we will continue to explore how this interaction and cargo transfer are preferentially facilitated. On one hand, we aim to gain a deeper understanding of the role of ether lipids. On the other, we are investigating purely biophysical parameters, such as membrane curvature and asymmetry, to better understand their relationship with interaction and transfer. This will help clarify the key physical parameters involved.

Maybe the authors could add few sentences about their perspectives.

MAJOR COMMENTS

The authors agreed to modify figure 2B and correct the caption of Figure s4 but these modifications are not included in the revised version.

· The authors refer to Fig. S3 when talking about two mobile populations and a bimodal model used to interpret the data (and obtain the population fractions). However, such Figure does not show HDL particles localization, neither refers (in the caption) to two mobile fractions. Maybe the authors could include another supplementary movie recorded on one the PSLB shown in Fig S3. Moreover, they do not comment on the reasons for the presence of two populations.

Response by the authors: In Figure S3, we present the raw data underlying the analysis shown in Figure 2B. These data pertain to lipid mixtures of fluid lipids (DOPC) and gel-phase lipids (DPPC and variants of ether lipids with different chain lengths), which are well-known to induce phase separation (Phase Transition Temperature, DOI: 10.1063/1.3258077, 10.1016/j.bbamem.2014.07.014). Consequently, these membrane systems develop into a two-phase mixture consisting of a mobile (DOPC) phase and a gel-phase-like immobile phase (DPPC, Ether Lipids). Figure S3 illustrates the localization of HDLap particles (middle row of images), which are predominantly found in the gel phase. Unfortunately, a video would not provide additional information, as this is a steady-state system with no diffusion observed. We hope this explanation adequately addresses the reviewer’s concerns.

I suggest the authors to comment on the presence of these two mobile populations and on their proportions (coefficient alpha) along the text, as they did in the caption of FigS2. Moreover, if I understood correctly, they should refer here to FigureS2 instead of S3.

· The authors claim “In summary, in a binary mixture with DOPC (18:1) ester instead of ether linkage (DPPC vs. both DietherPC lipids) as well as longer fatty acid chains (Diether PC (18:0) vs. Diether PC (16:0)) yields an enhanced interaction with HDL particles.”, however they show the opposite (HDL particles interact more with dieterPC (Fig 2A) and prefer partitioning in gel phase domains (Fig S3)). Indeed, in the conclusion (line 505) the authors comment that “lipids […] with an ester (i.e., DOPC (18:1)) instead of an ether (i.e., DietherPC (18:1)) glycerol linkage, exhibit a diminished interaction with HDL particles.”.

Response by the authors: The provided data also align with the visualization in Suppl. Fig. 3. However, the visual representation may not be entirely clear. Initially, we attempted to present the analysis as follows, but this likely caused some confusion. We are open to using different intensity scalings to improve visibility, though this approach could be misleading and may not result in a fair representation.

I think the figures reported in the original version of the paper are clear. However, from my understanding, HDL particle interaction is enhanced with shorter tails (Diether PC (16:0) vs. Diether PC (18:0)), and ether linkage (DietherPC lipids vs. DPPC), in binary mixtures with DOPC. If so, rephrasing is needed in line 409-410, since it is written the opposite.
